# The Clinical Features, Pathogenesis and Methotrexate Therapy of Chronic Chikungunya Arthritis

**DOI:** 10.3390/v11030289

**Published:** 2019-03-22

**Authors:** J. Kennedy Amaral, Peter C. Taylor, Mauro Martins Teixeira, Thomas E. “Tem” Morrison, Robert T. Schoen

**Affiliations:** 1Department of Infectious Diseases and Tropical Medicine, Federal University of Minas Gerais, Belo Horizonte, Minas Gerais 31270-901, Brazil; jkennedy-@hotmail.com; 2Nuffield Department of Orthopaedics, Rheumatology and Musculoskeletal Sciences, University of Oxford, Windmill Road, Oxford, OX3 7LDR, UK; peter.taylor@kennedy.ox.ac.uk; 3Department of Biochemistry and Immunology, Federal University of Minas Gerais, Belo Horizonte, Minas Gerais 31270-901, Brazil; mmtex.ufmg@gmail.com; 4Department of Immunology and Microbiology, University of Colorado School of Medicine, Aurora, CO 80045, USA; Thomas.morrison@udenver.edu; 5Section of Rheumatology, Allery and Immunology, Yale University School of Medicine, New Haven, CT 06510, USA

**Keywords:** chikungunya virus, chronic chikungunya arthritis, pathogenesis, methotrexate

## Abstract

Chikungunya fever (CHIKF) is an emerging viral infection that has spread widely, along with its *Aedes* vectors, throughout the tropics and beyond, causing explosive epidemics of acute illness and persistent disabling arthritis. The rheumatic symptoms associated with chikungunya virus (CHIKV) infection include polyarthralgia, polyarthritis, morning stiffness, joint edema, and erythema. Chronic CHIK arthritis (CCA) often causes severe pain and associated disability. The pathogenesis of CCA is not well understood. Proposed hypotheses include the persistence of a low level of replicating virus in the joints, the persistence of viral RNA in the synovium, and the induction of autoimmunity. In this review, we describe the main hypotheses of CCA pathogenesis, some of which support methotrexate (MTX) treatment which has been shown to be effective in preliminary studies in CCA.

## 1. Introduction

Chikungunya virus (CHIKV) is a small (60–70 nm, 12 Kb), single-stranded positive-sense RNA virus in the *Alphavirus* genus of the *Togaviridae* family [1,2]. Like many other arboviral diseases, chikungunya fever (CHIKF) is transmitted by mosquito vectors, primarily *Aedes aegypti* and *Aedes albopictus* [3]. CHIKF is an infectious febrile illness often accompanied by acute and chronic arthritis. The name “chikungunya” in the Makonde dialect refers to the stooped posture assumed by many patients who experience disabling joint pain [4].

CHIKV was isolated in 1952 during an outbreak in the Makonde Plateau, Tanzania (Africa), but CHIKF may be much older [5]. A disease similar to CHIKF was reported in Zanzibar in 1823, and since then, sporadic outbreaks have primarily been reported in Africa and parts of Asia. In the 21st century, CHIKF became a global epidemic. The disease occurred in Kenya in 2004 (500,000 cases), and in 2005, spread across countries in the Indian Ocean, including Reunion Island (266,000 cases) and India (1.4 million cases) [3,6,7]. In 2013, CHIKF was reported in the Caribbean island of Saint Martin and spread to 45 countries in the Americas (with more than 3 million cases) [6]. In Brazil alone, almost 500,000 cases of CHIKF have been reported since 2014 [8].

CHIKV infection typically occurs in two phases. In the first phase, 2 to 6 days after the mosquito bite, infected patients develop a high fever, arthralgia and arthritis, headache, maculopapular rash, and intense fatigue, often accompanied by anorexia, nausea, vomiting, and diarrhea [9]. In most reports, almost all of the infected patients became symptomatic, but high rates of asymptomatic infection have also been recorded in Thailand (47.1%) and Kenya (45.1%) with east/central/south African viral lineages [10]. Acute CHIKF lasts approximately 10 days [4]. Neurological complications, including encephalitis, optic neuritis, facial paralysis, sensorineural deafness, and Guillain–Barré syndrome, can occur in a variable proportion of patients [11]. Less frequently, CHIKF causes myocarditis, cardiac arrhythmia, severe sepsis, septic shock, and renal failure [3].

After the acute phase, the disease resolves in some patients, but 25%–40% develop chronic arthritis that includes musculoskeletal pain, arthralgia, or frank arthritis, lasting weeks or months after the acute attack. These symptoms can be relapsing or persistent and are unrelated to previous rheumatic disease, which is absent in the majority of patients [12,13].

CHIKF has become a global epidemic, causing acute febrile illness, chronic painful arthritis, and severe economic harm to the affected communities [4,7]. Unfortunately, at present, the only available treatment for acute infection is supportive—hydration, and use of opioid analgesics, paracetamol/acetaminophen and non-steroidal anti-inflammatory drugs (NSAIDs). Aspirin has been avoided due to the risk of bleeding, especially in patients who were co-infected with dengue or at risk of Reye’s syndrome. Similarly, corticosteroids have been avoided due to the concern for immunosuppression and potential exacerbation of the viral infection [4].

Despite the gaps in understanding of the mechanism of viral replication and pathogenesis of both acute and chronic disease, various antiviral therapeutics have been investigated [14,15]. These efforts include preclinical studies of the traditional antiviral compounds, synthesis of designer molecules, in silico high-throughput screening for existing products with efficacy against CHIKV, nucleic acid compounds, therapeutic monoclonal antibodies, and drugs that target host cell proteins [15]. Several compounds are in early clinical trials, including ribavirin, 6-azauridine, glycyrrhizin, and interferon. There has also been interest in vaccine development, including virus-like particle and measles-vectored vaccines that have reached phase 2 clinical trials [16]. However, at the present time, no antiviral therapies or licensed vaccines are available to lessen the impact of this epidemic [15,17].

As its pathogenesis is uncertain, there is no existing consensus as to how chronic chikungunya arthritis (CCA) should be treated. In addition to symptomatic treatment with NSAIDs, a variety of inflammation suppressing therapies, including corticosteroids, chloroquine (CHQ), hydroxychloroquine (HCQ), sulfasalazine (SSZ), methotrexate (MTX), and immune-modulating biologic agents including anti-TNF agents, B-lymphocyte depletion (rituximab), and interleukin-6 receptor inhibition (tocilizumab), have been used [13,18,19]. The treatment strategy has been empirical and no high-quality, controlled, randomized trials have assessed these interventions in CCA [19].

In this report, we discuss the clinical features and pathogenesis of CCA, including evidence for a persistent viral infection versus a post-infectious inflammatory disease. We considered MTX as a therapeutic option for CCA, based on the current understanding of its pathogenesis and evaluated the MTX studies that have been conducted in CCA [20]. Due to the severity and chronicity of CHIK arthritis, it is important to develop effective treatment [21]. We are optimistic about the use of MTX in CCA, based on favorable, albeit limited, clinical data [19,20,22].

## 2. Chronic Chikungunya Arthritis

The transition from CHIKF to CCA is variable. In many patients, arthrialgias/arthritis began at the onset of disease and is unremitting. In others, there is transient improvement after CHIKF followed by persisitent arthritis [4,12,23]. In several studies, 25% to 62% of patients had arthritic symptoms 18 months after the onset of CHIKF [23,24]. Even at the 36-month follow-up, the prevalence of arthritis has been reported to be as high as 60% [12,13,25]. Among 152 Colombian patients evaluated at 26 weeks after disease onset, persistent arthritis was found in 54%, morning stiffness in 49%, joint edema in 41%, and both polyarthralgia and morning stiffness in 38% [26]. In a Reunion Island study (2005–2006) of 88 patients, chronic arthritis occurred in 93%, 57%, and 47% at 3, 15, and 24 months, respectively [24]. In another Colombian cohort (2014–2015), persistent arthritis was less frequent, with 12% affected at the 18-month follow-up [23].

Rheumatic symptoms associated with CHIKV infection include polyarthralgia, polyarthritis, morning stiffness, joint edema, and joint redness. CHIK arthritis often causes severe pain and associated disability [21,27]. The most commonly affected joints are hands, knees, wrists, ankles, and shoulders [22,24,26,28]. For example, Borgherini et al. [24] reported the frequency of joints affected as hands (57%), knees (57%), wrists (50%), ankles (46%), and shoulders (45%). Mathew et al. [29] described the involvement of knees (83%), ankles (62%), and elbows (59%). The pattern of symmetric polyarthralgia/polyarthritis of large and small joints, especially in the hands, knees, shoulders, wrists, and ankles, has been frequently described [26,28,29,30,31].

A number of reports characterized CCA as “post-chikungunya chronic inflammatory rheumatism” (CIR-CHIK) [12,18,26,29,30,32]. Javelle et al. [13] described 159 patients who were symptomatic for at least 2 years, 112 of whom developed CIR-CHIK classified as mimicking four clinical patterns—rheumatoid arthritis (RA), seronegative spondyloarthritis (SpA), fibromyalgia (FM), or undifferentiated polyarthritis (UP), defined as the presence of inflammatory arthritis affecting more than four joints of greater than 6 weeks duration in the absence of an alternative diagnosis. Among these 112 patients, 33 fulfilled classification criteria for spondyloarthritis, 40 for RA, and 21 for undifferentiated polyarthritis [13,31]. In a group of 437 patients with CHIKF (India, 2011), 57% developed post-viral polyarthralgia, 22% inflammatory polyarthritis, and 19.5% tenosynovitis during a 15-month period [29]. Several of these patients’ diseases were reported to mimic RA and psoriatic arthritis (PsA) [13,30,33].

Similarly, Schilte et al. [25] studied 180 patients with CHIKV arthritis of 36 months duration in whom the hands, wrists, ankles, and knees were most affected. Around 60%–80% of the patients had intermittent arthritis, while arthritis was unremitting in 20%–40%. Among 173 patients with CHIKV evaluated at 27.5 months, Essackjee et al. [33] reported that 78.6% had persistent musculoskeletal symptoms and 5% met the American College of Rheumatology/European League Against Rheumatism (ACR/EULAR) criteria for RA.

Other reports have also focused on CCA as an “RA mimic”. Among the 203 patients with CHIKF who developed joint pain, 36% (34/94) met the ACR/EULAR criteria for RA [34]. In other reports, RA mimics have been even more frequent. In 39 Colombian patients with CHIKF, 90% developed arthritis that met the RA ACR/EULAR criteria [35]. In a small cohort of 10 American relief workers who developed CHIKF in Haiti, 8 of 10 fulfilled the ACR/EULAR criteria for seronegative RA, presenting with morning stiffness and polyarthritis, especially in the hands, wrists, feet, and ankles [36]. A typical example of symmetrical polyarthritis of the hands—reminiscent of RA—in one of our patients is shown in Figure 1. Another parallel to RA has been the finding of subchondral bone erosions, joint effusions, and joint thickening in patients with CCA [34,37].

Among reports of CCA patients whose disease mimics RA, the rates of rheumatoid factor (RF) and anti-cyclic citrullinated peptide antibody (CCP) positivity have varied widely. In an Indian study, 13 of 95 patients and 4 of 67 patients were respectively positive for RF and anti-CCP antibody [38]. In other reports, RF positivity has varied between 25% and 43%, and anti-CCP antibody positivity has been less frequent [39]. We observed a patient who developed RA symptoms 3 months after CHIKF who was positive for both RF and anti-CCP antibody. In addition, this patient developed neutropenia and splenomegaly within 1 year after the onset of her arthritis, establishing the diagnosis of Felty’s syndrome [40]. On the other hand, in a Colombian cohort of 109 CCA patients, 98% had no detectable RF or anti-CCP antibodies [12]. In the Haitian report, none of the 8 RA mimic patients had RF or anti-CCP antibody positivity [36], and of the 22 prospectively evaluated Reunion Island patients with long-term arthralgia, none were positive for anti-CCP antibody [13,31]. There may be two different groups of chronic arthritis patients being reported—CCA patients who are negative for RF and anti-CCP antibody, and another group that are more frequently positive for RF and anti-CCP antibody and actually suffering from two diseases, CHIKF followed by new onset of RA. 

Recently in our clinic, we evaluated a cohort of 50 Brazilian patients seen with CCA that we defined as arthritis/arthralgia, persisting for more than 3 months after the onset of CHIKF [41]. Our patients, 90% of whom were self-referred, experienced chronic pain and disability prior to evaluation in our clinic. The mean time between the onset of CHIKF and the first visit to us was 14.2 months, similar to the delays of 8 months to 2 years in treatment of CCA reported elsewhere [41]. Thirty of our patients (60%) had arthralgia while 20 patients (40%) also had arthritis, with clinically evident synovitis (Figure 1). Arthralgia was most common in the hands (56%), ankles (48%), and knees (44%). Arthralgia was polyarticular (>4 joints) (76%) or oligoarticular (2–4 joints) (24%). Of the patients with arthritis, all 20 had hand involvement. Other joints with arthritis included wrists in 16 (32%), ankles in 12 (24%), and ankles and knees in 9 (18%) patients. Overall, both large and small joints of the upper and lower extremities were affected. Morning stiffness, low back pain, and neck pain were reported by 3 (6%), 8 (16%), and 3 (6%) patients, respectively. The ACR criteria for RA were met by 11 (22%) of the patients, and another 7 (14%) met the criteria for FM [41].

We evaluated whether pre-existing rheumatic disease in our patients impacted the subsequent clinical expression of CCA and found that pre-existing rheumatic diseases could aggravate the pain in CCA patients. Similarly, among 159 cases of CHIK-CIR analyzed by Javelle et al. [12], 50 (31%) had previous “rheumatic and musculoskeletal disorders”, including 6 with “chronic inflammatory rheumatism” [12]. Sissoko [32] observed osteoarthritis in 38 of 147 (26%) patients who were followed for 15 months. In a small study, Zeana [42] reported preexisting rheumatic disease, including osteoarthritis, carpal tunnel syndrome, retrocalcaneal bursitis, and lateral epicondylitis. Essackjee [33] found that among 173 patients followed for more than 2 years after CHIKV infection, 27 (15.6%) had some pre-existing musculoskeletal disease. In our cohort, the rate of pre-existing rheumatic disease was significant (22 of 50; 44%), but there was no relationship between these conditions and the clinical expression of CHIK arthritis or the response to treatment [41].

Suggested risk factors for progression to long-term disease include female sex, age greater than 45, diabetes mellitus, hypertension, dyslipidemia, and previous rheumatic disease [43]. In addition, one study found that more severe initial CHIKV infection predicted a greater likelihood of long-term arthritis [44].

## 3. Replication Cycle of Chikungunya Virus

The transmission cycle of CHIKV requires the infection of female mosquitoes through a virus-containing blood meal and, following an appropriate extrinsic incubation period, transmission to another vertebrate host during subsequent feeding [1,4]. The ability of CHIKV to bind human cells and to replicate in cell cultures was documented, and the general features are similar to other alphaviruses [45,46,47]. CHIKV is internalized into human target cells mainly as a result of receptor-mediated endocytosis involving clathrin-dependent mechanisms, delivering the virus to endosomes from which the viral capsid is released into the cytosol [48,49,50].

The latter process is triggered by the low pH environment in the endosome that induces conformational changes in the viral E1 and E2 glycoproteins. These conformational changes result in exposure of the E1 fusion loop, which inserts into the host membrane and promotes fusion of the viral envelope and endosomal membrane. The synthesis of viral RNA occurs in the replication complexes found in the bulb-shaped invaginations of the plasma membrane, termed spherules, and includes the production of genomic positive-sense viral RNA, as well as a subgenomic positive-sense viral RNA, that encodes the structural polyprotein (capsid, E3, E2, 6K/TF, E1) from a negative-sense template. Following release of the capsid protein via autoproteolysis, the E1 and E2 glycoproteins are trafficked into the endoplasmic reticulum, transported through the Golgi, processed in the trans Golgi network, and finally transported to the plasma membrane where virus assembly and egress occur [49,51].

CHIKV is highly cytopathic in human cell cultures, and the infected cells rapidly undergo apoptosis [52]. Alphavirus replication strongly affects the fundamental processes of cellular physiology, with inhibition of cell transcription and translation, and redirection of cellular resources for the synthesis of viral proteins and viral genomes [52,53,54]. Studies have shown that CHIKV infection has been associated with extensive cell death and the release of high levels of infectious virus [46,54]. In human cell cultures, CHIKV elicits an autophagic response that promotes viral replication [55,56,57].

A study showed that the ability of CHIKV to induce apoptosis depends on the ability of the virus to replicate [55]. There are clues that the completion of the apoptotic process is an important element for the efficient propagation of the virus [58]. Inhibition of the apoptotic process by pan-caspase inhibitors (a family of intracellular cysteine proteases critical to several cellular functions, including apoptosis and inflammation) limits the number of CHIKV-infected cells [55,59]. A study showed that blood monocytes were the main targets of CHIKV during acute phase infection and that macrophages and B lymphocytes were also infected with the virus, contrary to previous reports [60]. CHIKV infection rapidly results in the induction of type I interferons (IFNs) and the production of proinflammatory cytokines as part of the innate cellular immune response [61,62].

## 4. Pathogenesis of Chronic Chikungunya Arthritis

There are intriguing similarities in the immunological phenotypes of peripheral blood mononuclear cells of patients with RA and CCA (CCA), but the pathobiology of CCA is not well understood. Proposed hypotheses include the persistence of a low level of replicating virus in the joints, the persistence of viral RNA in synovium, and the induction of autoimmunity [63].

### 4.1. Cytokine/Chemokine Responses

In some studies, CCA is associated with high levels of circulating IL-6, GM-CSF, IFN-α, and IL-17 [64,65]. In a study of the involvement of inflammatory cytokines and chemokines, plasma levels of IL-6 and GM-CSF were significantly higher in patients with persistent arthralgia compared with those who had recovered [66]. IL-6 is involved in the joint inflammation associated with RA and increases the production of cartilage-destroying enzymes. In addition to IL-6, CCA patients have other systemic markers of inflammation, such as the IFN-induced chemokines MIG/CXCL-9 and IP-10/CXCL-10, as well as proinflammatory cytokines including IL-1β, IL-1RA, CCL-2/MCP-1, CCL-3/MIP-1a, CCL-4/MIP-1b, and IL-12 [64]. MCP-1/CCL-2 is a major chemoattractant for monocytes and macrophages, and is strongly expressed during acute infection in humans and animal models [1,67].

In addition, the worsening severity of CHIKF has been associated with increased plasma levels of IL-1β and IL-6 and a reduced level in RANTES. As in RA, high levels of IL-1β may also mediate the development of abrupt and persistent arthralgia [68]. In a recent study, levels of all cytokines that were elevated in patients with CCA (IL1 -RA, IL-6, IL-8, MIP-1a, MIP-1b, and MCP-1) returned to control levels following disease resolution, indicating that elevations of these cytokines were specific for patients who develop chronic arthritis [65].

A study of the cytokine profile of patients in the acute and chronic phases of CHIKF identified dramatically elevated levels of IL-12 in chronic patients [64]. IL-12 is essential for the synthesis of Th1 lymphocytes, acting as a growth factor for natural killer (NK) cells, increasing their cytotoxic action [69]. In the synovial tissue analysis of patients with CCA, there was evidence of active monocyte/macrophage trafficking and several abnormal histological findings in the synovial membrane, including synovial lining hyperplasia, vascular proliferation, and infiltration of macrophages [64].

Most parenchymal cells express interleukin-17 receptors. Signaling through these receptors induces target cells to produce proinflammatory factors, such as IL-1, IL-6, IL-8, TNF, and matrix metalloproteinases capable of destroying the extracellular matrix and causing bone resorption [2,62,70]. This helps in understanding cases of bone erosion and joint damage in patients with CCA (Figure 2).

### 4.2. Cellular Response

An inefficient antiviral response due to disturbed immune cell function (NK, T cell, B cell, etc.) may be a possible reason for the persistence of the virus and/or chronic arthralgia [71]. For example, in one study, the expression of NKG2A and CD94 inhibitory receptors on natural killer (NK)/natural killer T cell (NKT) cells from chronic CHIK patients was elevated [71]. Another study that examined human synovial biopsies with CCA showed a high fraction of activated CD69 + CD4 + T cells, suggesting that these cells may contribute to chronic disease [64]. Indeed, a study reported the reduced frequency of NK-like T cells, lower expression of perforin + NK, and higher expression of TNF-α + NK-like T and IFN-γ + NK-like T cells as markers of chronic arthritic diseases [72].

### 4.3. Autoimmunity

Experiments using non-human primates and mice demonstrated that the RNA and CHIKV antigen remained detectable in musculoskeletal and other tissues for months after infection. In mice, the long-term persistence of viral RNA was associated with chronic inflammatory pathology and immune activation [73]. In the mouse model, chronic arthritis may be related to persistent, replicating, transcriptionally active CHIKV RNA [74]. In contrast, in an analysis of 33 patients 22 months after acute infection, no viral RNA or proteins were identified in the synovial fluid, suggesting that viral persistence may not be necessary for persistent arthritis [75]. The authors suggested that autoantigens or autoreactive lymphocytes could be present in the synovium or muscle tissue, contributing to the CCA.

Immunohistology on muscle biopsies from two CHIKV-infected patients with a myositic syndrome showed that viral antigens were found exclusively inside skeletal muscle progenitor cells; viral RNA has also been detected in synovial and muscle tissue biopsies collected from patients with CCA [51,63,76].

## 5. Methotrexate Therapy of Chronic Chikungunya Arthritis

To explain the anti-inflammatory properties of MTX, several mechanisms of action have been suggested, including the inhibition of purine and pyrimidine synthesis, suppression of transmethylation reactions with accumulation of polyamines, reduction of antigen-dependent T-cell proliferation, and promotion of adenosine release with adenosine-mediated suppression of inflammation [77,78]. In RA, the anti-inflammatory effects of MTX seem to be related to an extracellular increase in adenosine and its interaction with specific cell surface receptors. This is followed by the inhibition of purine and pyrimidine synthesis, mediated by the production of interleukin 8 (IL8)/CXCL8 by peripheral blood mononuclear cells (PBMC), the secretion of IL6 by human monocytes, and the inhibition of other proinflammatory cytokines, including IL-1, IL-2, IL-8, IL-12, and TNF-α [78,79]. In addition, MTX reduces the production of IL-4, IL-6, IL-13, TNF-α, interferon gamma (IFNγ), and granulocyte-macrophage colony-stimulating factor (GM-CSF), due to the de novo synthesis of purines and pyrimidines [80].

Studies in animal models have shown that a low dose of MTX promotes intracellular accumulation of 5-aminoimidazole-4-carboxamide ribonucleotide (AICAR), an intermediate in purine synthesis, and that the accumulation of AICAR is associated with increased adenosine release in inflammatory exudates; adenosine mediates the anti-inflammatory effects of MTX in animal models of both acute inflammation and adjuvant arthritis [77,81].

As CCA is similar to RA [26,30,36] and has a similar pathogenesis [2,62,63,64], the improvement of patients with CCA using MTX [20,41] could be due to the increase in adenosine, in addition to the decrease in levels of inflammatory interleukins.

A recent study revealed that MTX treatment is probably safe and does not affect the antiviral and inflammatory responses of primary human synovial fibroblasts (HSF). MTX alone did not increase the capacity of CHIKV to infect and replicate in HSF [82]. This fact is important because the immunomodulatory activity of MTX in the context of viral persistence has been of concern.

We conducted a systematic review to evaluate the efficacy and safety of MTX in CCA [20]. Among 131 possibly relevant available studies, six met our evaluation criteria. Of these, four were retrospective studies, one was an uncontrolled prospective study, and one was an unblinded randomized clinical trial comparing MTX monotherapy to MTX combined with sulfasalazine and hydroxychloroquine. Based on our review of published studies of MTX in CCA, we believe that (a) MTX treatment has been safe and (b) the evidence warrants high-quality randomized clinical trials of MTX in CCA [20].

As noted, we recently evaluated the effectiveness of MTX in pain reduction using a visual analog scale (VAS) (0–10) with higher values indicating more severe pain [41]. We treated 48 Brazilian CCA patients who were seen 14.2 (SD, 4.2) months after the onset of disease. Sixty percent of the patients had arthralgia. Forty percent also had arthritis. The patients received MTX 7.5 mg/week with folic acid, with dose escalations for refractory symptoms at 4 weeks. The final mean MTX dose was 9.2 (SD, 3.2) mg/week. The mean reductions in pain at 4 and 8 weeks, compared to baseline, were 4.3 (3.0) (*p* < 0.0001) and 4.5 (2.6) (*p* < 0.0001), respectively [41]

## 6. Conclusions

The reported improvement of CCA patients treated with MTX may be a pathway for understanding the uncertain pathogenesis of this emerging form of arthritis. The similarities in the clinical and pathogenic characteristics of CCA and RA make MTX an essential drug to evaluate in CCA. A better understanding of the treatment of CCA should provide further insights, not only in this disease, but in other forms of inflammatory arthritis, including RA.

## Figures and Tables

**Figure 1 viruses-11-00289-f001:**
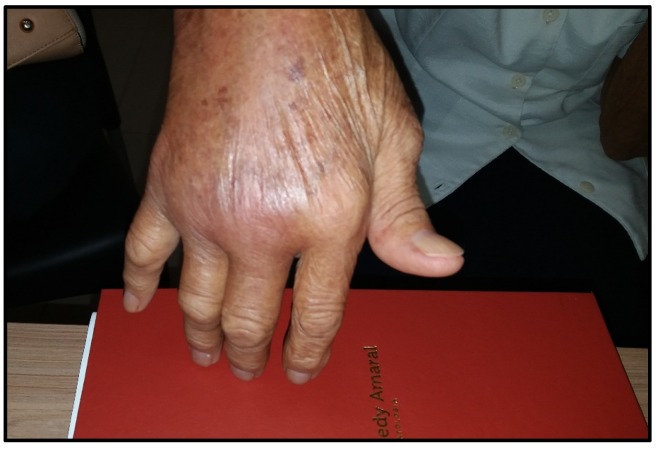
Woman, 82 years old, 2 years after CHIKV infection. Intense arthritis of metacarpophalangeal joints and wrist.

**Figure 2 viruses-11-00289-f002:**
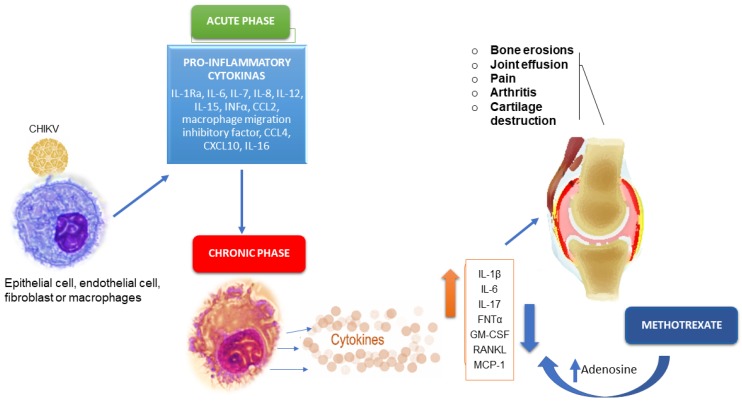
Cytokines that participate in the pathogenic process of acute and chronic phases of CHIKV infection and likely mechanism of action of MTX therapy.

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
