# Peer review of "The Clinical Features, Pathogenesis and Methotrexate Therapy of Chronic Chikungunya Arthritis"

_viruses, 2019, doi:10.3390/v11030289_

Round 1
Reviewer 1 Report
In the manuscript ‘The clinical features, pathogenesis and methotrexate therapy of chronic chikungunya arthritis’ by Amaral and colleagues, the authors review literature on chronic chikungunya arthritis, its symptoms, incidence, mechanism and methotrexate as a potential treatment option. While the subject is of high interest and a review on this topic welcome, the manuscript is a bit shallow at times and the authors tend to list off numbers/percentages in the text, without visualization or summarizing statements and perspectives. However, some important clinical aspects were summarized and the review could be informative to others in the field.
Major comments:
The authors refer twice to a ‘Figure 1’, and once to ‘Figure 2’, yet no figures were provided with the manuscript. Either these were omitted by accident, or the authors were referring to figures within the cited literature, which cannot be done in this manner and is confusing.
Generally, the authors describe many individual studies, including information on numbers of patients with selected symptoms. For these data, it would be useful to have a table or figure summarizing the clinical cases – it gets a bit hard to follow. A table/figure is not entirely necessary, but would make it easier to comprehend the most common symptoms of CCA.
Another major comment regards the methotrexate section. The authors include methotrexate treatment in the title and I was interested to read about it’s efficacy in treating CCA, however the section on methotrexate only goes into some detail on how it may or may not mediate its anti-inflammatory properties, without much information on how efficient it is in treating CHIKV induced persistent arthritis. In fact, the authors only mention its safety, but say nothing about its efficacy until the conclusion section where they just mention ‘evident improvement’.
Furthermore, the section ‘Replication cycle of chikungunya virus’ may be better suited as part of the introduction, or as a first section, since it disrupts the CCA section and the ‘pathogenesis of CCA’ section, which would flow better after one another.
Minor comments:
Affiliations: The affiliations are numbered, but the numbers are not provided behind each name.
References: References throughout the text are placed behind the period of each sentence, which is confusing. References should be added in front of the period of the sentence they are referring to, e.g. ‘… disabling joint pain [4].’
Line 38: only two examples are listed following ‘including’ and should thus be connected with an ‘and’ instead of a comma: ‘ …, including Reunion Island (266,000 cases) and India (1.4 million cases).’
Line 93: Might be useful to say ‘in another Colombian cohort’ since the authors previously mentioned a Colombian cohort in the same context.
Line 106: switch RA and rheumatoid arthritis – i.e. introduce the abbreviation in parentheses, not the explanation of the abbreviation in parentheses
Line 110: Since the authors are introducing a new study in the sentence starting ‘In a group of 437 patients …’, it would be good to mention year or location to separate it from the previous sentence.
Line 127: The authors refer to a ‘Figure 1’, which is not present. When checking, I realized the authors are probably referring to Figure 1 in the cited reference (?). This is not clear here and should either be shown in a figure or referred to differently.
Line 144: typo, should be ‘arthralgia’ instead of arthrialgia’
Line 149: No Figure 1 was provided with this manuscript???
Line 264: The authors state: ‘This is followed by inhibition, mediated by…’ without stating what is inhibited. Please be specific here.
Line 275: ‘RA arthritis’ is redundant and should be just ‘RA’.
Line 283: I believe the authors meant ‘CCA patients treated with MTX’ here.
Author Response
The authors refer twice to a ‘Figure 1’, and once to ‘Figure 2’, yet no figures were provided with the manuscript. Either these were omitted by accident, or the authors were referring to figures within the cited literature, which cannot be done in this manner and is confusing.
Response: Figures 1 and 2 are included at the end of the text.
Generally, the authors describe many individual studies, including information on numbers of patients with selected symptoms. For these data, it would be useful to have a table or figure summarizing the clinical cases – it gets a bit hard to follow. A table/figure is not entirely necessary, but would make it easier to comprehend the most common symptoms of CCA.
Response: We have not included an additional summary table/figure
Another major comment regards the methotrexate section. The authors include methotrexate treatment in the title and I was interested to read about it’s efficacy in treating CCA, however the section on methotrexate only goes into some detail on how it may or may not mediate its anti-inflammatory properties, without much information on how efficient it is in treating CHIKV induced persistent arthritis. In fact, the authors only mention its safety, but say nothing about its efficacy until the conclusion section where they just mention ‘evident improvement’.
Response: We have added additional information on methotrexate efficacy (lines 283-296).
Furthermore, the section ‘Replication cycle of chikungunya virus’ may be better suited as part of the introduction, or as a first section, since it disrupts the CCA section and the ‘pathogenesis of CCA’ section, which would flow better after one another.
Response: We have not moved the section “Replication cycle of the chikungunya virus.
Minor comments:
Affiliations: The affiliations are numbered, but the numbers are not provided behind each name.
References: References throughout the text are placed behind the period of each sentence, which is confusing. References should be added in front of the period of the sentence they are referring to, e.g. ‘… disabling joint pain [4].’
Line 38: only two examples are listed following ‘including’ and should thus be connected with an ‘and’ instead of a comma: ‘ …, including Reunion Island (266,000 cases) and India (1.4 million cases).’
Line 93: Might be useful to say ‘in another Colombian cohort’ since the authors previously mentioned a Colombian cohort in the same context.
Line 106: switch RA and rheumatoid arthritis – i.e. introduce the abbreviation in parentheses, not the explanation of the abbreviation in parentheses
Line 110: Since the authors are introducing a new study in the sentence starting ‘In a group of 437 patients …’, it would be good to mention year or location to separate it from the previous sentence.
Line 127: The authors refer to a ‘Figure 1’, which is not present. When checking, I realized the authors are probably referring to Figure 1 in the cited reference (?). This is not clear here and should either be shown in a figure or referred to differently.
Line 144: typo, should be ‘arthralgia’ instead of arthrialgia’
Line 149: No Figure 1 was provided with this manuscript???
Line 264: The authors state: ‘This is followed by inhibition, mediated by…’ without stating what is inhibited. Please be specific here.
Line 275: ‘RA arthritis’ is redundant and should be just ‘RA’.
Line 283: I believe the authors meant ‘CCA patients treated with MTX’ here.
Response: We have revised the article to incorporate all minor comment recommendations
Reviewer 2 Report
Amaral and authors present an updated discussion on a complicated medical issue of chronic chikungunya arthritis. This manuscript will be of interest to the chikungunya field and viral arthritis in general.
Some suggestions for improvements, transparency and clarity
Some references are after the full-stop while some are before. For improved read-through, perhaps put all numbered references before the full-stop.
Line 39, full stop missing
Line 40 "with more than 3 million cases"
Neurological complications following chikungunya virus infection is highly variable and dependent on the study location, design of the study (cross-sectional, case-control etc), co-morbidities etc. Therefore presenting a single statistic of "up to 25% of patients", is not representative of these complications. And the reference used in the following sentence is not easily accessible. Perhaps use a systematic review of chikungunya neurological complications such as Mehta et al., 2018, Rev Med Virol; 28(3): e1978.
Line 68, vaccine development including particle-like should be replaced with virus-like particles for accuracy
As it is still not really well defined what is the mechanism behind chikungunya virus chronicity, some additional statements may improve transparency. e.g. No infectious virus has been recovered during the chronic phase of chikungunya infection either in humans of preclinical models. In addition, a discussion of what is known in mouse models on chikungunya virus chronicity may also be useful. e.g. there is evidence to suggest that during the chronic phase of chikungunya infection in wildtype mice, chronic arthritic disease is a consequence of persistent, replicating and transcriptionally active CHIKV RNA (Poo et al., PLoS Negl Trop Dis. 2014;8(12):e3354)
The methotrexate section could use some expanding. Some statements related to the economic costs of chronic chikungunya arthritis will be useful to provide justification for why an alternative treatment is needed. Some background into methotrexate such as indication use and costs (e.g. FDA approved chemotherapeutic drug, also used for rheumatoid arthritis etc) would also be helpful. This paragraph could also include some discussion about other studies testing other drugs against chronic chikungunya infection (anakinra, sulfalazine).
Author Response
Some references are after the full-stop while some are before. For improved read-through, perhaps put all numbered references before the full-stop.
Line 39, full stop missing
Line 40 "with more than 3 million cases"
Neurological complications following chikungunya virus infection is highly variable and dependent on the study location, design of the study (cross-sectional, case-control etc), co-morbidities etc. Therefore presenting a single statistic of "up to 25% of patients", is not representative of these complications. And the reference used in the following sentence is not easily accessible. Perhaps use a systematic review of chikungunya neurological complications such as Mehta et al., 2018, Rev Med Virol; 28(3): e1978.
Line 68, vaccine development including particle-like should be replaced with virus-like particles for accuracy
As it is still not really well defined what is the mechanism behind chikungunya virus chronicity, some additional statements may improve transparency. e.g. No infectious virus has been recovered during the chronic phase of chikungunya infection either in humans of preclinical models. In addition, a discussion of what is known in mouse models on chikungunya virus chronicity may also be useful. e.g. there is evidence to suggest that during the chronic phase of chikungunya infection in wildtype mice, chronic arthritic disease is a consequence of persistent, replicating and transcriptionally active CHIKV RNA (Poo et al., PLoS Negl Trop Dis. 2014;8(12):e3354)
The methotrexate section could use some expanding. Some statements related to the economic costs of chronic chikungunya arthritis will be useful to provide justification for why an alternative treatment is needed. Some background into methotrexate such as indication use and costs (e.g. FDA approved chemotherapeutic drug, also used for rheumatoid arthritis etc) would also be helpful. This paragraph could also include some discussion about other studies testing other drugs against chronic chikungunya infection (anakinra, sulfalazine).
Response:
We have corrected several suggested stylistic issues.
We have added discussion based on the suggested references regarding neurological manifestations and viral persistence. We have also expanded the discussion on MTX efficacy (lines 283-296) as also requested by Review 1.
I hope these revisions are satisfactory